# Momentum Contrastive Autoencoder: Using Contrastive Learning for Latent Space Distribution Matching in WAE

## Abstract

Wasserstein autoencoder (WAE) shows that matching two distributions is equivalent to minimizing a simple autoencoder (AE) loss under the constraint that the latent space of this AE matches a pre-specified prior distribution. This latent space distribution matching is a core component of WAE, and a challenging task. In this paper, we propose to use the contrastive learning framework that has been shown to be effective for self-supervised representation learning, as a means to resolve this problem. We do so by exploiting the fact that contrastive learning objectives optimize the latent space distribution to be uniform over the unit hyper-sphere, which can be easily sampled from. We show that using the contrastive learning framework to optimize the WAE loss achieves faster convergence and more stable optimization compared with existing popular algorithms for WAE. This is also reflected in the FID scores on CelebA and CIFAR-10 datasets, and the realistic generated image quality on the CelebA-HQ dataset.

## 1 Introduction

The main goal of generative modeling is to learn a good approximation of the underlying data distribution from finite data samples, while facilitating an efficient way to draw samples. Popular algorithms such as variational autoencoders (VAE, Kingma & Welling (2013); Rezende et al. (2014)) and generative adversarial networks (GAN, Goodfellow et al. (2014)) are theoretically-grounded models designed to meet this goal. However, they come with some challenges. For instance, VAEs suffer from the posterior collapse problem (Chen et al., 2016; Zhao et al., 2017; Van Den Oord et al., 2017), and a mismatch between the posterior and prior distribution (Kingma et al., 2016; Tomczak & Welling, 2018; Dai & Wipf, 2019; Bauer & Mnih, 2019). GANs are known to have the mode collapse problem (Che et al., 2016; Dumoulin et al., 2016; Donahue et al., 2016) and optimization instability (Arjovsky & Bottou, 2017) due to their saddle point problem formulation.

Wasserstein autoencoder (WAE) Tolstikhin et al. (2017) proposes a general theoretical framework that can potentially avoid some of these challenges. They show that the divergence between two distributions is equivalent to the minimum reconstruction error, under the constraint that the marginal distribution of the latent space is identical to a prior distribution. The core challenge of this framework is to match the latent space distribution to a prior distribution that is easy to sample from. Tolstikhin et al. (2017) investigate GANs and maximum mean discrepancy (MMD, Gretton et al. (2012)) for this task and empirically find that the GAN-based approach yields better performance despite its instability. Existing research has tried to address this challenge (Kolouri et al., 2018; Knop et al., 2018) (see section 2 for a discussion).

This paper aims to design a generative model to address the latent space distribution matching problem of WAEs. To do so, we make a simple observation that allows us to use the contrastive learning framework. Contrastive learning achieves state-of-the-art results in self-supervised representation learning tasks (He et al., 2020; Chen et al., 2020) by forcing the latent representations to be 1) augmentation invariant; 2) distinct for different data samples. It has been shown that the contrastive learning objective corresponding to the latter goal pushes the learned representations to achieve maximum entropy over the unit hyper-sphere (Wang & Isola, 2020). We observe that applying this contrastive loss term to the latent representation of an AE therefore matches it to the uniform

distribution over the unit hyper-sphere. Due to the use of the contrastive learning approach, we call our algorithm Momentum Contrastive Autoencoder (MoCA). Our contributions are as follows:

1. we address the fundamental algorithmic challenge of Wasserstein auto-encoders (WAE), viz the latent space distribution matching problem, which involves matching the marginal distribution of the latent space to a prior distribution. We achieve this by making the observation that the contrastive term in the recent contrastive learning framework implicitly achieves this precise goal. This is also our novelty.

2. we show that our proposal of using the contrastive learning framework to optimize the WAE loss achieves faster convergence and more stable optimization compared with existing popular algorithms for WAE.

3. we perform a thorough ablation analysis of the impact of the hyper-parameters introduced by the contrastive learning framework, on the performance and behavior of WAE.

## 2  RELATED WORK

There has been a considerable amount of research on autoencoder based generative modeling. In this paper we focus on Wasserstein autoencoders (WAE). Nonetheless, we discuss other autoencoder methods for completeness, and then focus on prior work that aim at achieving the WAE objective.

**AE based generative models**: One of the earliest model in this category is the de-noising autoencoder (Vincent et al., 2008). Bengio et al. (2013b) show that training an autoencoder to de-noise a corrupted input leads to the learning of a Markov chain whose stationary distribution is the original data distribution it is trained on. However, this results in inefficient sampling and mode mixing problems (Bengio et al., 2013b; Alain & Bengio, 2014).

Variational autoencoders (VAE) (Kingma & Welling, 2013; Rezende et al., 2014) overcome these challenges by maximizing a variational lower bound of the data likelihood, which involves a KL term minimizing the divergence between the latent's posterior distribution and a prior distribution. This allows for efficient approximate likelihood estimation as well as posterior inference through ancestral sampling once the model is trained. Despite these advantages, followup works have identified a few important drawbacks of VAEs. The VAE objective is at the risk of posterior collapse – learning a latent space distribution which is independent of the input distribution if the KL term dominates the reconstruction term (Chen et al., 2016; Zhao et al., 2017; Van Den Oord et al., 2017). The poor sample qualities of VAE has been attributed to a mismatch between the prior (which is used for drawing samples) and the posterior (Kingma et al., 2016; Tomczak & Welling, 2018; Dai & Wipf, 2019; Bauer & Mnih, 2019). Dai & Wipf (2019) claim that this happens due to mismatch between the AE latent space dimension and the intrinsic dimensionality of the data manifold (which is typically unknown), and propose a two stage VAE to remedy this problem. VQ-VAE (Oord et al., 2017) take a different approach and propose a discrete latent space as an inductive bias in VAE.

Ghosh et al. (2019) observe that VAEs can be interpreted as deterministic autoencoders with noise injected in the latent space as a form of regularization. Based on this observation, they introduce deterministic autoencoders and empirically investigate various other regularizations.

Similar to our work, the recently proposed DC-VAE (Parmar et al., 2021) also uses contrastive loss for generative modeling. However, the objective resulting from their version of instance discrimination estimates the log likelihood function rather than the WAE objective (which estimates Wasserstein distance). Also, they integrate the GAN loss to the instance discrimination version of VAE loss.

There has been research on AEs with hyperspherical latent space, that we use in our paper. Davidson et al. (2018) propose to replace the Gaussian prior used in VAE with the Von Mises-Fisher (vMF) distribution, which is analogous to the Gaussian distribution but on the unit hypersphere.

**WAE**: Tolstikhin et al. (2017) make the observation that the optimal transport problem can be equivalently framed as an autoencoder objective under the constraint that the latent space distribution matches a prior distribution. They experiment with two alternatives to satisfy this constraint in the form of a penalty – MMD (Gretton et al., 2012) and GAN (Goodfellow et al., 2014)) loss, and they find that the latter works better in practice. Training an autoencoder with an adversarial loss was also proposed earlier in adversarial autoencoders (Makhzani et al., 2015).

There has been research on making use of sliced distances to achieve the WAE objective. For instance, Kolouri et al. (2018) observe that Wasserstein distance for one dimensional distributions have a closed form solution. Motivated by this, they propose to use sliced-Wasserstein distance, which involves a large number of projections of the high dimensional distribution onto one dimensional spaces which allows approximating the original Wasserstein distance with the average of one dimensional Wasserstein distances. A similar idea using the sliced-Cramer distance is introduced in Knop et al. (2018).

Patrini et al. (2020) on the other hand propose a more general framework which allows for matching the posterior of the autoencoder to any arbitrary prior of choice (which is a challenging task) through the use of the Sinkhorn algorithm (Cuturi, 2013). However, it requires differentiating through the Sinkhorn iterations and unrolling it for backpropagation (which is computationally expensive); though their choice of Sinkhorn algorithm for latent space distribution matching allows their approach to be general.

## 3 MOMENTUM CONTRASTIVE AUTOENCODER

We present the proposed algorithm in this section. We begin by restating the WAE theorem that connects the autoencoder loss with the Wasserstein distance between two distributions. Let $X \sim P_X$ be a random variable sampled from the real data distribution on $\mathcal{X}$, $Z \sim Q(Z|X)$ be its latent representation in $\mathcal{Z} \subseteq \mathbb{R}^d$, and $\hat{X} = g(Z)$ be its reconstruction by a deterministic decoder/generator $g : \mathcal{Z} \to \mathcal{X}$. Note that the encoder $Q(Z|X)$ can also be deterministic in the WAE framework, and we let $f(X) \stackrel{dist}{=} Q(Z|X)$ for some deterministic $f : \mathcal{X} \to \mathcal{Z}$.

**Theorem 1.** *(Bousquet et al., 2017; Tolstikhin et al., 2017) Let $P_Z$ be a prior distribution on $\mathcal{Z}$, let $P_g = g\#P_Z$ be the push-forward of $P_Z$ under $g$ (i.e. the distribution of $\hat{X} = g(Z)$ when $Z \sim P_Z$), and let $Q_Z = f\#P_X$ be the push-forward of $P_X$ under $f$. Then,*

$$W_c(P_X, P_g) = \inf_{Q:Q_Z=P_Z} \mathbb{E}_{\substack{X \sim P_X \\ Z \sim Q(Z|X)}} [c(X, g(Z))] = \inf_{f:(f\#P_X)=P_Z} \mathbb{E}_{X \sim P_X} [c(X, g(f(X)))] \quad (1)$$

*where $W_c$ denotes the Wasserstein distance for some measurable cost function $c$.*

The above theorem states that the Wasserstein distance between the true ($P_X$) and generated ($P_g$) data distributions can be equivalently computed by finding the minimum (w.r.t. $f$) reconstruction loss, under the constraint that the marginal distribution of the latent variable $Q_Z$ matches the prior distribution $P_Z$. Thus the Wasserstein distance itself can be minimized by jointly minimizing the reconstruction loss w.r.t. both $f$ (encoder) and $g$ (decoder/generator) as long as the constraint is met.

In this work, we parameterize the encoder network $f : \mathcal{X} \to \mathbb{R}^d$ such that latent variable $Z = f(X)$ has unit $\ell_2$ norm. Our goal is then to match the distribution of this $Z$ to the uniform distribution over the unit hyper-sphere $\mathcal{S}_d = \{z \in \mathbb{R}^d : \|z\|_2 = 1\}$. To do so, we study the so-called "negative sampling" component of the contrastive loss used in self-supervised learning,

$$L_{neg}(f; \tau, K) = \mathbb{E}_{\substack{x \sim P_X \\ \{x_i^-\}_{i=1}^K \sim P_X}} \left[ \log \frac{1}{K} \sum_{j=1}^K e^{f(x)^T f(x_j^-)/\tau} \right] \quad (2)$$

Here, $f : \mathcal{X} \to \mathcal{S}_d$ is a neural network whose output has unit $\ell_2$ norm, $\tau$ is the temperature hyper-parameter, and $K$ is the number of samples (another hyper-parameter). Theorem 1 of Wang & Isola (2020) shows that for any fixed $t$, when $K \to \infty$,

$$\lim_{K \to \infty} (L_{neg}(f; \tau, K) - \log K) = \mathbb{E}_{x \sim P_X} \left[ \log \mathbb{E}_{x^- \sim P_X} \left[ e^{f(x)^T f(x^-)/\tau} \right] \right] \quad (3)$$

Crucially, this limit is minimized *exactly* when the push-forward $f\#P_X$ (i.e. the distribution of the latent random variable $Z = f(X)$ when $X \sim P_X$) is uniform on $\mathcal{S}_d$. Moreover, even the Monte Carlo approximation of Eq. 2 (with mini-batch size $B$ and some $K$ such that $B \le K < \infty$)

$$L_{neg}^{MC}(f; \tau, K, B) = \frac{1}{B} \sum_{i=1}^B \log \frac{1}{K} \sum_{j=1}^K e^{f(x_i)^T f(x_j)/\tau} \quad (4)$$

---

**Algorithm 1** PyTorch-like pseudocode of Momentum Contrastive Autoencoder algorithm

---

```
# Enc_q, Enc_k: encoder networks for query and key. Their outputs are L2 normalized
# Dec: decoder network
# Q: dictionary as a queue of K randomly initialized keys (dxK)
# m: momentum
# lambda: regularization coefficient for entropy maximization
# tau: logit temperature

for x in data_loader: # load a minibatch x with B samples
    z_q = Enc_q(x) # queries: Bxd
    z_k = Enc_k(x).detach() # keys: Bxd, no gradient through keys
    x_rec = Dec(z_q) # reconstructed input

    # positive logits: Bx1
    l_pos = bmm(z_q.view(B,1,d), z_k.view(B,d,1))

    # negative logits: BxK
    l_neg = mm(z_q.view(B,d), Q.view(d,K))

    # logits: Bx(1+K)
    logits = cat([l_pos, l_neg], dim=1)

    # compute loss
    labels = zeros(B) # positive elements are in the 0-th index
    L_con = CrossEntropyLoss(logits/tau, labels) # contrastive loss maximizing entropy of z_q
    L_rec = ((x_rec - x) ** 2).sum() / B # reconstruction loss
    L = L_rec + lambda * L_con # momentum contrastive autoencoder loss

    # update Enc_q and Dec networks
    L.backward()
    update(Enc_q.params)
    update(Dec.params)

    # update Enc_k
    Enc_k.params = m * Enc_k.params + (1-m) * Enc_q.params

    # update dictionary
    enqueue(Q, z_k) # enqueue the current minibatch
    dequeue(Q) # dequeue the earliest minibatch
```

---

bmm: batch matrix multiplication; mm: matrix multiplication; cat: concatenation.
enqueue appends $Q$ with the keys $z_k \in \mathbb{R}^{B \times d}$ from the current batch; dequeue removes the oldest $B$ keys from $Q$

---

is a consistent estimator (up to a constant) of the entropy of $f \# P_X$ called the redistribution estimate (Ahmad & Lin, 1976). This follows if we notice that $k(x_i; \tau, K) := \frac{1}{K} \sum_{j=1}^{K} e^{f(x_i)^T f(x_j)/\tau}$ is the un-normalized kernel density estimate of $f(x_i)$ using the i.i.d. samples $\{x_j\}_{j=1}^{K}$, so $-L_{neg}^{MC}(f; \tau, K, B) = -\frac{1}{B} \sum_{i=1}^{B} \log k(x_i; \tau, K)$ (Wang & Isola, 2020). So minimizing $L_{neg}$ (and importantly $L_{neg}^{MC}$) maximizes the entropy of $f \# P_X$.

Tolstikhin et al. (2017) attempted to enforce the constraint that $f \# P_X$ and $P_Z$ were matching distributions by regularizing the reconstruction loss with the MMD or a GAN-based estimate of the divergence between $f \# P_X$ and $P_Z$. By letting $P_Z$ be the uniform distribution over the unit hyper-sphere $\mathcal{S}_d$, the insights above allow us to instead minimize the much simpler regularized loss

$$L(f, g; \lambda, \tau, B, K) = \frac{1}{B} \sum_{i=1}^{B} \|x_i - g(f(x_i))\|_2^2 + \lambda L_{neg}^{MC}(f; \tau, K, B) \tag{5}$$

**Training**: For simplicity, we will now use the notation $\mathrm{Enc}(\cdot)$ and $\mathrm{Dec}(\cdot)$ to respectively denote the encoder and decoder network of the autoencoder. Further, the $d$-dimensional output of $\mathrm{Enc}(\cdot)$ is $\ell_2$ normalized, i.e., $\|\mathrm{Enc}(x)\|_2 = 1 \, \forall x$. Based on the theory above, we aim to minimize the loss $L(\mathrm{Enc}, \mathrm{Dec}; \lambda, \tau, B, K)$, where $\lambda$ is the regularization coefficient, $\tau$ is the temperature hyper-parameter, $B$ is the mini-batch size, and $K \geq B$ is the number of samples used to estimate $L_{neg}$.

In practice, we propose to use the momentum contrast (MoCo, He et al. (2020)) framework to implement $L_{neg}$. Let $\mathrm{Enc}_t$ be parameterized by $\theta_t$ at step $t$ of training. Then, we let $\mathrm{Enc}'_t$ be the same encoder parameterized by the exponential moving average $\tilde{\theta}_t = (1 - m) \sum_{i=1}^{t} m^{t-i} \theta_i$. Letting $x_1, \ldots, x_K$ be the $K$ most recent training examples, and letting $t(j) = t - \lfloor j/B \rfloor$ be the time at

which $x_j$ appeared in a training mini-batch, we replace $L_{neg}^{MC}$ at time step $t$ with

$$L_{MoCo} = \frac{1}{B} \sum_{i=1}^{B} \log \frac{1}{K} \sum_{j=1}^{K} \exp \left( \frac{\text{Enc}_t(x_i)^T \text{Enc}'_{t(j)}(x_j)}{\tau} \right) \quad - \frac{1}{B} \sum_{i=1}^{B} \frac{\text{Enc}_t(x_i)^T \text{Enc}'_t(x_i)}{\tau} \quad (6)$$

This approach allows us to use the latent vectors of inputs outside the current mini-batch without re-computing them, offering substantial computational advantages over other contrastive learning frameworks such as SimCLR (Chen et al., 2020). Forcing the parameters of $\text{Enc}'$ to evolve according to an exponential moving average is necessary for training stability, as is the second term encouraging the similarity of $\text{Enc}_t(x_i)$ and $\text{Enc}'_t(x_i)$ (so-called "positive samples" in the terminology of contrastive learning). Note that we do not use any data augmentations in our algorithm, but this similarity term is still non-trivial since the networks $\text{Enc}_t$ and $\text{Enc}'_t$ are not identical. Pseudo-code of our final algorithm, which we call Momentum Contrastive Autoencoder (MoCA), is shown in Algorithm 1 (pseudo-code style adapted from He et al. (2020)). Finally, in all our experiments, inspired by Grill et al. (2020) we set the exponential moving average parameter $m$ for updating the $\text{Enc}'$ network at the $t^{th}$ iteration as $m = 1 - (1 - m_0) \cdot (\cos(\pi t/T) + 1)/2$, where $T$ is the total number of training iterations, and $m_0$ is the base momentum hyper-parameter.

**Inference**: Once the model is trained, the marginal distribution of the latent space (i.e. the push-forward $\text{Enc}\#P_X$) should be close to a uniform distribution over the unit hyper-sphere. We can therefore draw samples from the learned distribution as follows: we first sample $z \sim \mathcal{N}(0, I)$ from the standard multivariate normal distribution in $\mathbb{R}^d$ and then generate a sample $x_g := \text{Dec}(z/\|z\|_2)$.

## 4 EXPERIMENTS

We conduct experiments on CelebA (Liu et al., 2015), CIFAR-10 (Krizhevsky et al., 2009), CelebA-HQ (Karras et al., 2018) datasets, and synthetic datasets using our proposed algorithm as well as existing algorithms that implement the WAE objective. Unless specified otherwise, for CIFAR-10 and CelebA datasets, we use two architectures: A1: the architecture from Tolstikhin et al. (2017), which is commonly used as a means to fairly compare against existing methods; and A2: a ResNet-18 based architecture with much fewer parameters. For CelebA-HQ, we use a variant of ResNet-18 with 6 residual blocks instead of 4 for both the encoder and decoder. The remaining architecture and optimization details are provided in appendix A.

### 4.1 CONVERGENCE SPEED AND OBJECTIVE ESTIMATION

**Latent Space**: We try to evaluate how well the contrastive term in our objective addresses the problem of matching the marginal distribution $Q_Z = \text{Enc}\#P_X$ of the autoencoder latent space to the prior distribution $P_Z$, viz, the uniform distribution on the unit hyper-sphere. To systematically study the approximation quality and convergence rate in the latent space, we design a synthetic task where we remove the reconstruction loss, and set the objective to exclusively match the latent distribution with the prior distribution (thus the decoder network is ignored). Specifically, for all the methods we investigate, the reconstruction error term is removed and only the regularization term aimed at latent space distribution matching is used. Note that this is still a non-trivial objective.

To this end, we generate 100 dimensional synthetic data points as our dataset (1000 samples) which are fed through a 2 layer MLP to get their corresponding latent representations (128 dimensional). We then use different algorithms for matching the latent space distribution with a prior distribution (chosen to be the uniform distribution over the unit hyper-sphere in 128 dimensions). We use the Sinkhorn loss, Sliced Wasserstein Distance (SWD) and Maximum Mean Discrepancy (MMD) as baselines. For all algorithms, we use Adam optimizer with learning rate $1e - 3$ and train for 80 epochs on the synthetic data. In order to study how well the different algorithms match the latent space marginal distribution to the prior, during the training process, we measure the distance between the latent representations of the synthetic input data and random points sampled from the uniform distribution over the unit hyper-sphere in 128 dimensions. We use Sliced Wasserstein Distance (SWD) to measure this distance[1]. These SWD estimates are shown in figure 1. As evident, the contrastive objective converges significantly faster.

---

[1]Note that we use SWD both as an evaluation metric, as well as one of the baseline loss function for distribution matching in this experiment. To explain why SWD as an objective performs worse than some of

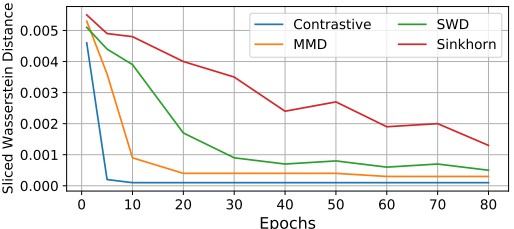 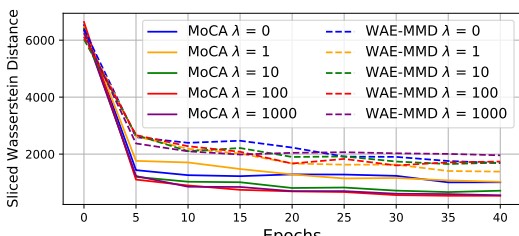

Figure 1: Systematic analysis of convergence speed for various objectives for matching the marginal distribution of the latent space to a prior, without the reconstruction term. Contrastive loss is faster than existing methods at this task. SWD measured in latent space.

Figure 2: Convergence speed and quality of objective approximation for MoCA vs WAE-MMD (baseline) for various values of their respective regularization coefficient $\lambda$ on CIFAR-10. SWD measured in image space.

**Image Space**: We measure how well our algorithm approximates the WAE objective compared with baseline. Since the MMD loss outperforms other baseline methods in the above experiments in the latent space, we only compare our algorithm with WAE-MMD (original WAE algorithm from Tolstikhin et al. (2017)), and empirically study the comparative convergence rate of these two algorithms. We train MoCA and WAE-MMD with their respective regularization coefficients $\lambda \in \{0, 1, 10, 100, 1000\}$ on CIFAR-10 using architecture A2. Starting at initialization (epoch=0), at every 5 epochs during training, we estimate the Sliced Wasserstein distance (SWD) between the test set images and the samples generated by each algorithm at the corresponding epoch. We use the code provided in [2] to compute SWD, whose implementation is aimed specifically at measuring the SWD between 2 image datasets. The results are shown in figure 2 (all hyper-parameters were chosen to be identical for both algorithms for fairness). The figure shows that larger $\lambda$ values in general result in smaller SWD, and SWD estimates decrease with epochs for MoCA, confirming that our proposed algorithm indeed minimizes the Wasserstein Distance. Additionally, looking at the comparative convergence rate of MoCA and WAE-MMD, we can make three observations:

1. Faster convergence: at epoch 5, the SWD for MoCA is already below 1800 compared with that of WAE-MMD (above 2300) across all values of the regularization coefficient $\lambda$. This shows that MoCA converges faster than WAE-MMD.

2. Better approximation: the final SWD values of MoCA are significantly lower than those of WAE-MMD across the corresponding $\lambda$ values.

3. Stable optimization: larger values of $\lambda$ result in smaller values of SWD for MoCA but this is not always true for WAE-MMD (see $\lambda = 1000$ for WAE-MMD).

## 4.2 LATENT SPACE BEHAVIOR

**Isotropy**: We qualitatively investigate the behavior of MoCA in terms of latent space distribution matching, i.e., how closely $Q_Z = \text{Enc}\#P_X$ of the autoencoder latent space matches the prior distribution $P_Z$. Since the encoder is parameterized to output unit $\ell_2$ norm vectors, we only need to evaluate how close $Q(Z)$ is to being isotropic. As a computationally efficient proxy, we compute the singular value decomposition (SVD) of the latent representation corresponding to 10,000 randomly sampled images from the training set. We use SVD because the spread of singular values of the latent space samples is indicative of how isotropic the latent space is. If the singular values of all the singular vectors are close to each other, the latent space distribution is more isotropic.

In this experiment, we train MoCA on CIFAR-10. We compute SVD of latent representation for models trained with different values of the regularization coefficient $\lambda \in \{0, 500, 1000, 3000\}$. Larger $\lambda$ is designed to increase the entropy of the latent space to better match it to the uniform distribution. As Figure 3 shows, for models trained with larger $\lambda$, the singular values (and therefore the latent

---

the other objectives, we hypothesize that its gradients are worse. A distant, but related example is classification tasks, where we care about accuracy, but accuracy as a loss function does not provide any gradients.

[2] https://github.com/koshian2/swd-pytorch

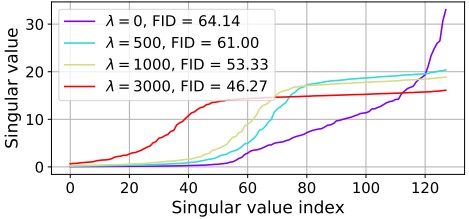
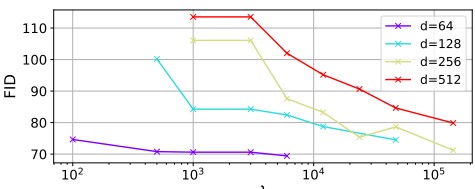

Figure 3: SVD of latent representation $\in$ $\mathbb{R}^{128}$ for models trained with various values of $\lambda$. Larger $\lambda$ results in more uniform singular values, i.e., closer to uniform distribution, and lower (better) FID for generated samples.

Figure 4: The interplay between the latent dimension $d$ of the MoCA network and the regularization coefficient $\lambda$. Larger $d$ requires significantly larger $\lambda$ to achieve comparable FID scores for the generated sample quality.

distribution) become more uniform. Corresponding FID scores on generated samples reflect this effect since models trained with larger $\lambda$ have lower (better) FID scores.

**Latent space dimensionality** $d$ **and regularization coefficient** $\lambda$ Real data often lies on a low dimensional ($d_0 < n$) manifold that is embedded in a high dimensional ($n$) space (Bengio et al., 2013a). Autoencoders attempt to map the probability distribution of the data to a designated prior distribution in a latent space of of dimension $d$, and vice versa. However, if there is a significant mismatch in the dimension $d_0$ of the true data manifold and the latent space's dimension $d$, learning a mapping between the two becomes impossible (Dai & Wipf, 2019). This results in many "holes" in the learned latent space which do not correspond to the training data distribution.

Given the importance of this problem, we study how the latent dimension $d$ influences the quality of samples generated by our used by our model MoCA. We also simultaneously analyze the influence of the regularization coefficient $\lambda$, since the value of $\lambda$ enforces how much we want the mapped latent distribution to be close to the uniform distribution on the unit hyper-sphere.

For this experiment, we use $d \in \{64, 128, 256, 512\}$ and study the value of $\lambda$ on a wide range on the log scale between 100 and 144,000 (in some cases). Due to the large number of experiments in this analysis, we train each configuration until the epoch reconstruction loss (mean squared error) reaches 50. For this reason the FID scores are much higher than the fully trained models reported in other experiments (where reconstruction loss reaches ~25). The results are shown in Figure 4. We find that for $d = 64$ the performance is quite stable across different $\lambda$ values. However, as $d$ is set to larger values, a significantly larger value of $\lambda$ is required to reach a similar FID score. We hypothesize that this happens because in Eq. 6, the dot product of the two encoding vectors is more likely to be orthogonal in a higher dimensional space, which suggests that the value of the contrastive regularization term becomes smaller compared with the reconstruction term in Eq. 5 (considering all other factors fixed). To compensate for this, a larger regularization coefficient would be required.

**t-SNE Visualization of Latent Space**: We present a qualitative comparison between the latent representation representations learned by Hyperspherical VAE ($\mathcal{S}$-VAE, Davidson et al. (2018)) and MoCA since both algorithms are aimed at learning a latent representation that is embedded on the unit hypersphere. For this experiment, we train both algorithms on the training set of the MNIST dataset. Once the models are trained, we compute the latent representation for the test set using both models. These latent representations are projected to a 2 dimensional space using T-SNE for visualization. The projections are shown in Figure 7 in appendix. We find that despite its simplicity, MoCA learns perceptually distinguishable class clusters similar to $\mathcal{S}$-VAE. Experimental details are provided in appendix B.

### 4.3 IMAGE GENERATION QUALITY

Qualitatively, we visualize random samples from our trained models on all the datasets. Figure 5 (rows 1-2) contains random samples from CelebA-HQ. The faces in these images look reasonably realistic. Figure 6 (rows 5-6) contains random samples from CIFAR-10 and CelebA, as well as reconstructions (rows 1-2) of images from these datasets. Most generated samples look realistic

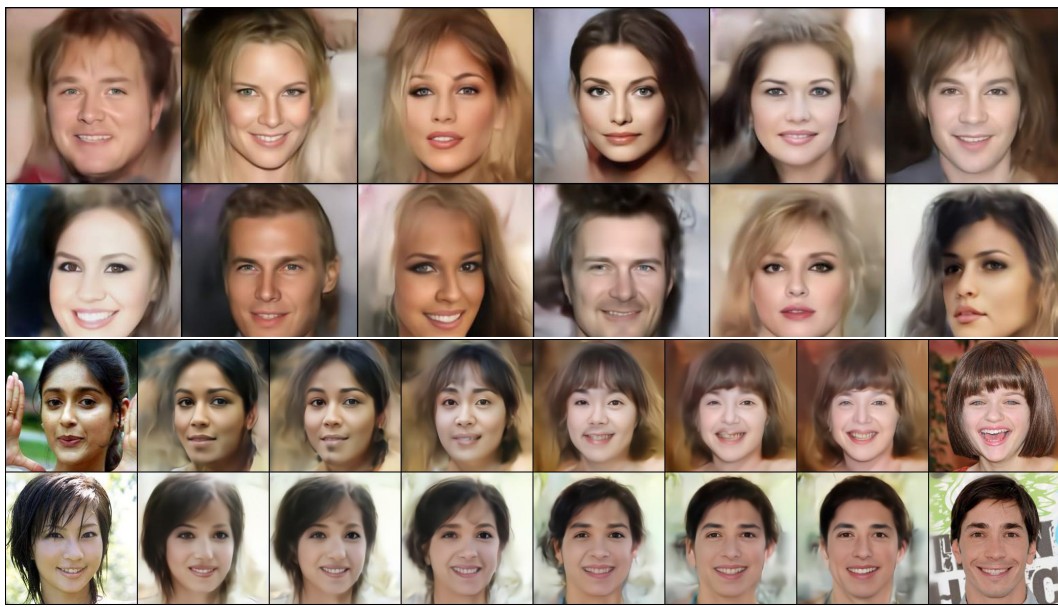

Figure 5: Random samples (rows 1-2) from a model trained with MoCA on CelebA-HQ, and that model's interpolations (rows 3-4) between images in latent space. The leftmost and rightmost columns of rows 3-4 are the original images from the test set of CelebA-HQ which we are interpolating.

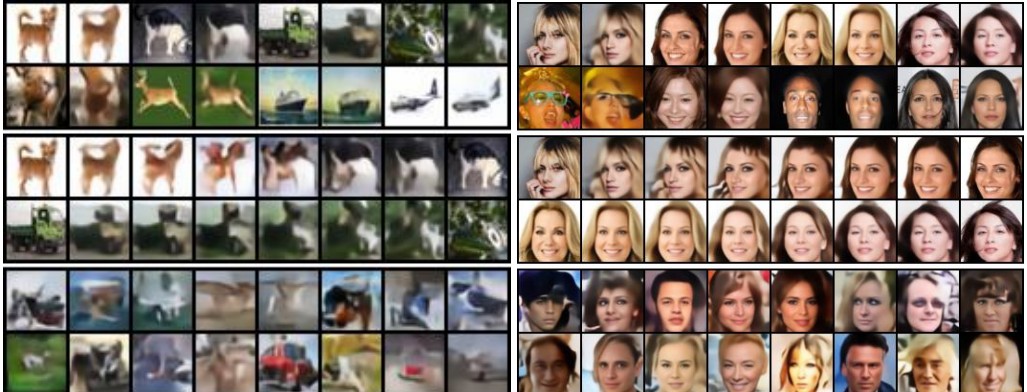

Figure 6: **Left**: CIFAR-10. **Right**: CelebA. Rows 1-2 show original image (odd column) and its reconstruction (even column). Rows 3-4 show model's interpolation between two test images in latent space. The leftmost and rightmost columns of rows 3-4 are the original images from the corresponding test set. Rows 5-6 show random samples drawn from a trained model.

(especially for the two CelebA datasets), and the reconstructions perceptually match the original input in most cases.

We also present latent space interpolations between images from the test set of CelebA-HQ in Figure 5 (rows 3-4). We present latent space interpolations for CIFAR-10 and CelebA in Figure 6 (rows 3-4). For these interpolations, we compute the latent vectors $z = \text{Enc}(x)$ and $z' = \text{Enc}(x')$ for two images $x$ and $x'$, let $z_\alpha = \alpha z + (1 - \alpha)z'$ for some $0 \leq \alpha \leq 1$, and then generate the interpolated image $\hat{x}_\alpha = \text{Dec}(z_\alpha/\|z_\alpha\|_2)$. These latent space interpolations show that our algorithm causes the generator to learn a smooth function from the unit hyper-sphere $\mathcal{S}_d$ to image space, and moreover, almost all intermediate samples look quite realistic.

We present additional random samples, image reconstructions, and latent space interpolations for these three datasets in appendix E.

| Data\Model | WAE-MMD | WAE-GAN | Sinhorn AE | SWAE | CWAE | MoCA-A1 | MoCA-A2 |
|---|---|---|---|---|---|---|---|
| CelebA | 55 | 42 | 56 | 79 | 49.69 | 48.43 | 44.59 |

Table 1: Comparison of MoCA with existing baselines using FID (lower is better) on CelebA dataset. MoCA-A1 uses an architecture similar to the one used in WAE (Tolstikhin et al., 2017), while MoCA-A2 uses a ResNet-18 based architecture.

### 4.4 QUANTITATIVE COMPARISON

For quantitative analysis we report the Fréchet Inception Distance (FID) score (Heusel et al., 2017). We compare the performance of MoCA with other existing algorithms that try to achieve the WAE objective and AE algorithms with hyper-spherical latent space, because these two class of algorithms are directly related to our proposal. Specifically, we compare with WAE-MMD and WAE-GAN (Tolstikhin et al., 2017), Sinkhorn autoencoder (Patrini et al., 2020) (with hyperspherical latent space), sliced Wasserstein autoencoder (SWAE, Kolouri et al. (2018)) and Cramer-Wold autoencoder (CWAE, Knop et al. (2018)). All numbers are cited from their respective papers.

Results are shown in table 1. We find that MoCA outperforms all of the methods except WAE-GAN, which uses GAN objective for matching the latent space distribution to the prior. Experiments comparing MoCA with WAE-MMD on CIFAR-10 can be found in appendix C.

### 4.5 ABLATION ANALYSIS AND IMPACT OF HYPER-PARAMETERS INTRODUCED BY THE CONTRASTIVE LEARNING LOSS

Unlike most existing autoencoder based generative models, our proposal of using the contrastive learning framework, specifically momentum contrastive learning (He et al., 2020) due to its computational efficiency compared to its competitor Chen et al. (2020), introduces a number of hyper-parameters in addition to the regularization coefficient $\lambda$. Therefore, it is important to shed light on their behavior during the training process of MoCA. This section explores how these various hyper-parameters impact the quality of generated samples. To keep the analysis tractable and quantitative, we use the Fréchet Inception Distance (FID) score to evaluate the performance of the trained models.

Due to lack of space, we discuss these experiments in appendix D, and summarize the results here. In appendix D.1, we study the impact of the regularization coefficient $\lambda$ used in MoCA, on the reconstruction loss achieved at the end of training, and find that larger values of $\lambda$ help reduce the reconstruction loss even more. In appendix D.2, we discuss how the regularization coefficient $\lambda$ should be scaled depending on the input image size, and we find that its optimal value scales linearly with input size. In appendix D.3, we discuss the impact of the momentum hyper-parameter used in the contrastive algorithm, and show that its value must be closer to 1 for good performance. In appendix D.4, we discuss the impact of the temperature hyper-parameter used in the contrastive algorithm, and show that its value must be smaller than 1 for good performance, and discuss possible reasons. Finally, in appendix D.5, we study the impact of the size of dictionary used in the contrastive algorithm, and find the the performance remains stable across a wide range of sizes.

## 5 CONCLUSION

We propose a novel algorithm for addressing the latent space distribution matching problem of Wasserstein autoencoders (WAE) called Momentum Contrastive Autoencoder (MoCA). The main idea behind MoCA is to use the contrastive learning framework to match the autoencoder's latent space marginal distribution with the uniform distribution on the unit hyper-sphere. We show that using the contrastive learning framework to estimate the WAE objective achieves faster convergence and more stable optimization compared with existing popular algorithms for WAE. We perform a thorough ablation analysis and study the impact of the hyper-parameters introduced in the WAE framework due to our proposal of using the contrastive learning, and discuss how to set their values.

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

APPENDIX

## A    TRAINING AND EVALUATION DETAILS

We ran all our experiments in Pytorch 1.5.0 (Paszke et al., 2019). We used 8 V100 GPUs on google could platform for our experiments.

**Datasets**:

**CIFAR-10** contains 50k training images and 10k test images of size $32 \times 32$.

**CelebA** dataset contains a total of $\sim$203k $64 \times 64$ images divided into $\sim$180k training images and $\sim$20k test images. The images were pre-processed by first taking 140x140 center crops and then resizing to 64x64 resolution.

**CelebA-HQ** contains a total of $\sim$30k $1024 \times 1024$ images. We resized these images to $256 \times 256$ and split the dataset into $\sim$27k training images and $\sim$3k test images.

**Architecture and optimization**:

The network architecture A1 is identical to the CNN architecture used in Tolstikhin et al. (2017) except that we use batch norm (Ioffe & Szegedy, 2015) in every layer (similar to Ghosh et al. (2019)), and the latent dimension is 128 for the CelebA dataset. This architecture roughly has around 38 million parameters. We found that using 64 dimensions for CelebA in this architecture prevented the reconstruction loss from reaching small values.

The encoder of the network architecture A2 is a modification of the standard ResNet-18 architecture He et al. (2016) in that the first convolutional layer has filters of size $3 \times 3$, and the final fully connected layer has latent dimension 128. The decoder architecture is a mirrored version of the encoder with upsampling instead of downsampling. Additionally, the final convolutional layer uses an upscaling factor of 1 for CIFAR-10 and 2 for CelebA. The architecture roughly has around 24 million parameters.

Both A1 and A2 were trained on CIFAR-10 with MoCA hyperparameters $K = 30000, \tau = 0.05, m_0 = 0.99$. A1 used $\lambda = 2000$ while A2 used $\lambda = 100$. A1 was trained for 40 epochs while A2 was trained for 100 epochs using the Adam optimizer with batch size 64, and learning rate 0.001. The learning rate was decayed by a factor of 2 after 60 epochs for A2.

Both A1 and A2 were trained on CelebA with MoCA hyperparameters $K = 60000, \tau = 0.05, m_0 = 0.99$. A1 used $\lambda = 1000$ while A2 used $\lambda = 100$. Both models were trained for 200 epochs using the Adam optimizer with batch size 64, and learning rate 0.001 decayed by a factor of 2 every 60 epochs.

During our experiments, we found that the choice of hyper-parameters $\tau$ and $m$ was stable across the two architectures and datasets and they were chosen based on our ablation studies. The value of $K$ was decided based on the size of the dataset (CelebA being larger than CIFAR-10 in our case). Finally, we found that the value of $\lambda$ was generally subjective to the dataset and architecture being used. We typically ran a grid search over $\lambda \in \{100, 1000, 3000, 6000\}$.

The images in Figure 5 (CelebA-HQ $256 \times 256$) were generated using a variant of the ResNet-18 architecture. The base ResNet-18 architecture has 4 residual blocks, each containing 2 convolutional layers and an additional convolutional layer which spatially downsamples its input by a factor of $2 \times 2$. For the encoder, we use the same architecture, but with 6 blocks (to downsample a $256 \times 256$ image to $4 \times 4$, which we then flatten and project into the latent space). The decoder is a mirrored version of the encoder, but with de-convolution upsampling layers instead of downsampling layers. The latent space of this architecture is 128 dimensional. We train this model with MoCA hyperparameters $\lambda = 20000, K = 30000, \tau = 1, m_0 = 0.99$. The model was trained for 1000 epochs using the Adam optimizer with batch size 64, and learning rate 0.002 (decayed by a factor of 2 every 40 epochs until epoch 400).

**Quantitative evaluation**:

In all our experiments, FID was always computed using the test set of the corresponding dataset. We always use 10,000 samples for computing FID.

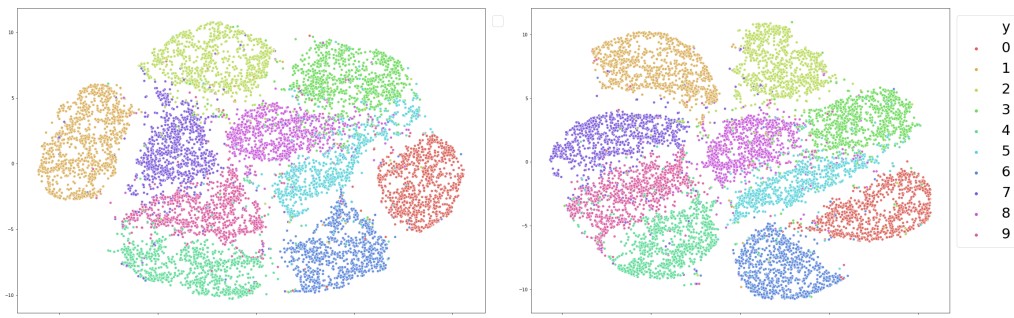

Figure 7: **Left**: MoCA. **Right**: Hyperspherical VAE ($\mathcal{S}$-VAE). T-SNE projections of autoencoder latent representation of MNIST test set. Both algorithms learn a hyperspherical latent space embedding. The colors correspond to the 10 digit classes in MNIST. Despite the simplicity of MoCA over $\mathcal{S}$-VAE, both algorithms are comparable at implicitly learning perceptually distinguishable class clusters.

| Data\Model | WAE-MMD | MoCA-A1 | MoCA-A2 |
|---|---|---|---|
| CIFAR-10 | 80.9 | 61.49 | 54.36 |

Table 2: Comparison of MoCA with WAE-MMD on CIFAR-10 dataset using FID (lower is better).

# B   EXPERIMENTAL DETAILS FOR EXPERIMENTS COMPARING MoCA WITH $\mathcal{S}$-VAE

For the T-SNE experiment, we train both algorithms using Adam optimizer for 25 epochs with a fixed learning rate 0.001 (other hyper-parameters take the default Pytorch Adam values). For MoCA, we used $\lambda = 5$, $K = 10,000$, $\tau = 0.99$. Both algorithms use the same MLP architecture with 3 hidden layer encoder and 2 hidden layer decoder, and ReLU activations. In both models, the hidden layer dimensions was 128 for all layers and the latent space dimension was 6. The latent space visualizations are shown in figure 7.

# C   ADDITIONAL QUANTITATIVE RESULTS

Experiments comparing MoCA with WAE-MMD on CIFAR-10 can be found in table 2.

# D   ABLATION ANALYSIS

For this section, unless specified otherwise, we use the CelebA dataset with the ResNet-18 autoencoder architecture (architecture A2 in the previous section), $\tau = 1$, $m_0 = 0.999$, $d = 128$, $\lambda = 3000$, $K = 60000$. For optimization, we use Adam with learning rate 0.001, batch size 64, drop this learning rate by half every 60 epochs, and train for a total of 200 epochs. All other optimization hyper-parameters are set to the default Pytorch values.

| size | $32 \times 32$ | $64 \times 64$ | $256 \times 256$ |
|---|---|---|---|
| $\lambda^\star$ | 100 | 2000 | 20000 |

Table 3: The optimal value of $\lambda$ scales linearly with input size. We consider $\lambda$ between 100 and 50000 and report the value $\lambda^\star$ that achieves the best FID score for each input size.

| $m_0$ | 0 | 0.9 | 0.99 | 0.999 |
|---|---|---|---|---|
| FID | 86.57 | 98.97 | 47.96 | 47.46 |

Table 4: Smaller base momentum $m_0$ causes model performance to degrade significantly. Performance is measured using the FID score (lower is better). Note that we use the cosine schedule described in section 3 in all these experiments.

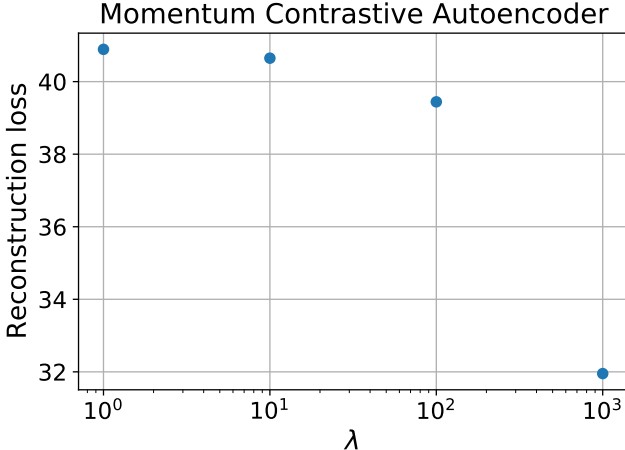

Figure 8: Impact of regularization coefficient $\lambda$ on reconstruction loss at the end of training. Larger values of $\lambda$ do not interfere with the reconstruction loss, rather help achieve lower reconstruction loss.

### D.1 RECONSTRUCTION LOSS VS $\lambda$

We study the impact of the regularization coefficient $\lambda$ used during the training of MoCA, on reconstruction loss achieved at the end of training. To provide some context, in VAEs, a trade-off emerges between the reconstruction loss and the KL term (regularization) because if KL is fully minimized, the posterior $p(z|x)$ becomes a Normal distribution and hence independent of the input $x$. Therefore, too strong a regularization worsens the reconstruction loss in VAEs. However, such a trade-off does not exist in WAE, in which the regularization aims the make the marginal distribution of the latent space close to the prior. We see this effect in the experiments shown in figure 8 for CIFAR-10 dataset using the A2 architecture. We find that larger values of $\lambda$ in fact help reduce the reconstruction loss even more.

### D.2 CHOOSING $\lambda$ GIVEN INPUT SIZE

An important consideration when selecting the regularization coefficient $\lambda$ is the relative scale of the reconstruction loss $\|x_i - g(f(x_i))\|_2^2$ and contrastive loss.

We show that the optimal value of the regularization weight $\lambda$ scales linearly with input size. For this experiment, we downscale CelebA-HQ ($256 \times 256$) to $64 \times 64$ and $32 \times 32$, and we study the impact of $\lambda$ for the different input sizes. We construct the models for generating $d \times d$ images by removing $\log_2(256/d)$ of the 6 residual blocks from the encoder/decoder of the base model used for CelebA-HQ ($256 \times 256$) (described in appendix A), as each block downsamples/upsamples the image by a factor of $2 \times 2$. We train all models for 400 epochs using the Adam optimizer with batch size 64 and learning rate 0.002 (decayed by a factor of 2 every 40 epochs).

For each case, we report the value of $\lambda \in \{100, 200, 500, 1000, 2000, 5000, 10000, 20000, 50000\}$ that achieves the best FID score for each image size. We find that the optimal value of $\lambda$ is roughly proportional to the number of pixels in the input (Table 3).

The full data for this experiment (which support the claims of Table 3) can be found in Figure 9. We consider $\lambda \in \{100, 200, 500, 1000, 2000, 5000, 10000, 20000, 50000\}$ for each image size (except $256 \times 256$, as quality rapidly deteriorates when $\lambda < 5000$). Please note that the absolute FID scores are not comparable between different image sizes! Rather, we emphasize the relative trends.

### D.3 IMPORTANCE OF MOMENTUM $m$

We study the impact of the contrastive learning hyper-parameter $m$ on the generated sample quality. $m$ is the exponential moving average hyper-parameter used for updating the parameters of the momentum encoder network $\text{Enc}_k$. $m$ is typically kept to be close to 1 for training stability (He et al., 2020). We confirm this intuition for our generative model as well in Table 4. We use the base

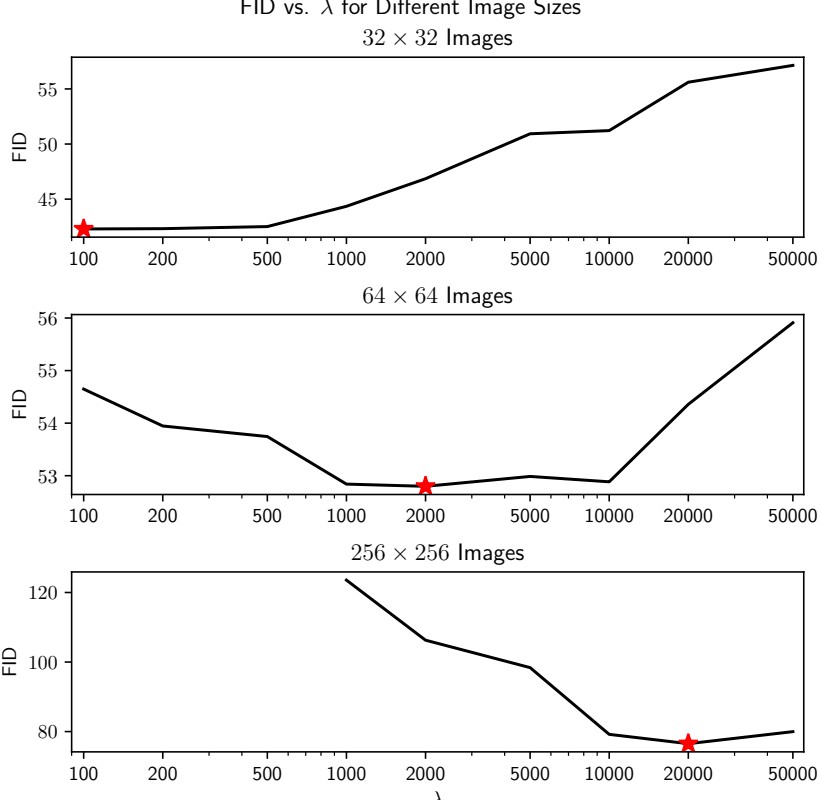

Figure 9: Impact of regularization weight $\lambda$ on FID score based on input size. Optimal values $\lambda^\star$ (labeled with a red star) scale linearly with input size. Note that absolute FID scores are not comparable between different image sizes! This figure focuses on relative trends.

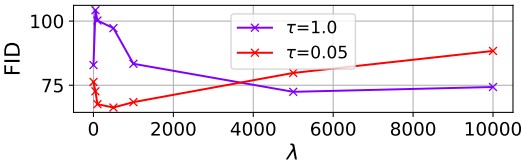

Figure 10: Impact of $\tau$ on optimal choice of $\lambda$ and best FID for generated samples. Optimal $\lambda$ is lower for lower $\tau$. Best FID is better for lower $\tau$. This suggests that entropy is maximized more accurately when $\tau$ is smaller.

value $m_0 \in \{0, 0.9, 0.999\}$. Note that we use the cosine schedule to compute the value of $m$ every iteration (as discussed in section 3), which makes $m$ increase from the base value $m_0$ to 1 over the course of training. We find that FID scores are much worse when $m$ is not close to 1.

## D.4 IMPORTANCE OF TEMPERATURE $\tau$

Based on the discussion below Eq. 3, the negative term in the contrastive loss essentially estimates the entropy of the latent space distribution due to its equivalent kernel density estimation (KDE) interpretation (Eq. 4). Therefore, the temperature hyper-parameter $\tau$ used in the contrastive loss acts as the smoothing parameter of this KDE and controls the granularity of the estimated distribution. Thus for larger temperature, the estimated distribution becomes smoother and the entropy estimation becomes poor, which should result in poor quality of generated samples. Additionally, for larger $\tau$,

| $K$ | 100 | 5000 | 10,000 | 30,000 | 60,000 | 120,000 |
|-----|-----|------|--------|--------|--------|---------|
| FID | 84.93 | 48.45 | 46.56 | 47.31 | 46.68 | 49.94 |

Table 5: The effect of dictionary size $K$ on the quality of samples measured using the FID score (lower is better). The performance is largely stable across different values of $K$ unless $K$ is too small.

intuitively, a larger $\lambda$ should be needed in order to push the KDE samples apart from one another. We confirm these intuitions in Figure 10. Note that due to the large number of experiments in this analysis, we train each configuration until 50 epochs (explaining the inferior FID values).

### D.5 EFFECT OF $K$

Since we use the momentum contrastive framework, it would be useful to understand how the dictionary size $K$ affects the quality of generative model learned. The dictionary $Q$ contains the negative samples in the contrastive framework which are used to push the latent representations away from one another, encouraging the latent space to be more uniformly distributed. We therefore expect a small $K$ would be bad for achieving this goal. Our experiments in Table 5 confirm this intuition. We use $K \in \{100, 5000, 10000, 30000, 60000, 120000\}$. We find that the FID score is stable and small across the various values of $K$ chosen, except for $K = 100$, for which FID is much worse.

## E ADDITIONAL QUALITATIVE RESULTS

Figures 11, 12, and 13 respectively depict additional randomly sampled images, reconstructions, and latent space interpolations for CelebA-HQ $256 \times 256$. Figures 12 and 13 are generated by the same model used to generate Figure 5, while Figure 11 is generated by an earlier checkpoint of that model (selected for best visual quality).

Figures 14, 16, and 18 respectively depict additional randomly sampled images, reconstructions, and latent space interpolations for CIFAR-10 using the same model that achieved the FID score of 54.36 in table 1.

Figures 15, 17, and 19 respectively depict additional randomly sampled images, reconstructions, and latent space interpolations for CelebA using the same model that achieved the FID score of 44.59 in table 1.

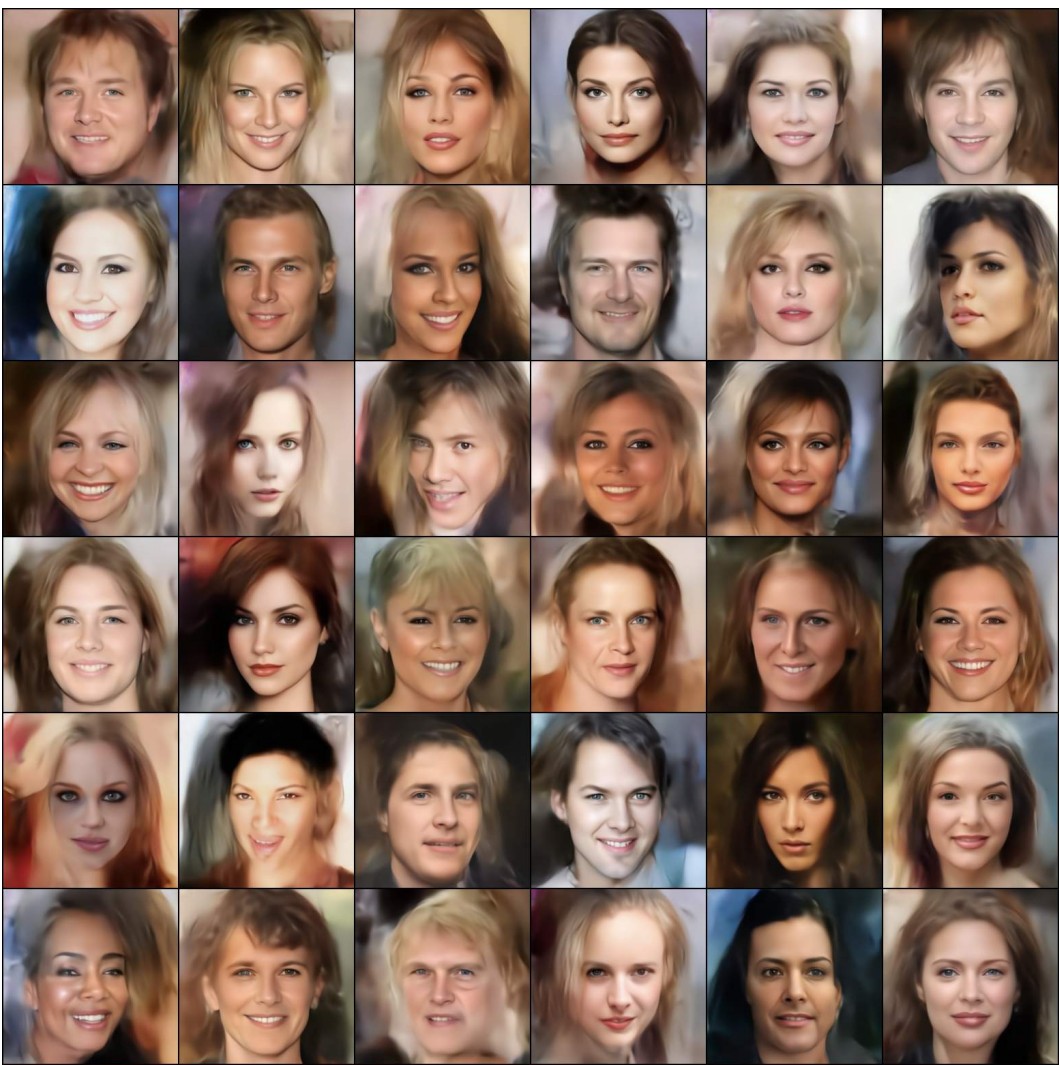

Figure 11: Random samples generated by a model trained (as described in appendix A on CelebA-HQ $256 \times 256$ for 850 epochs. Model checkpoint picked based on best visual quality.

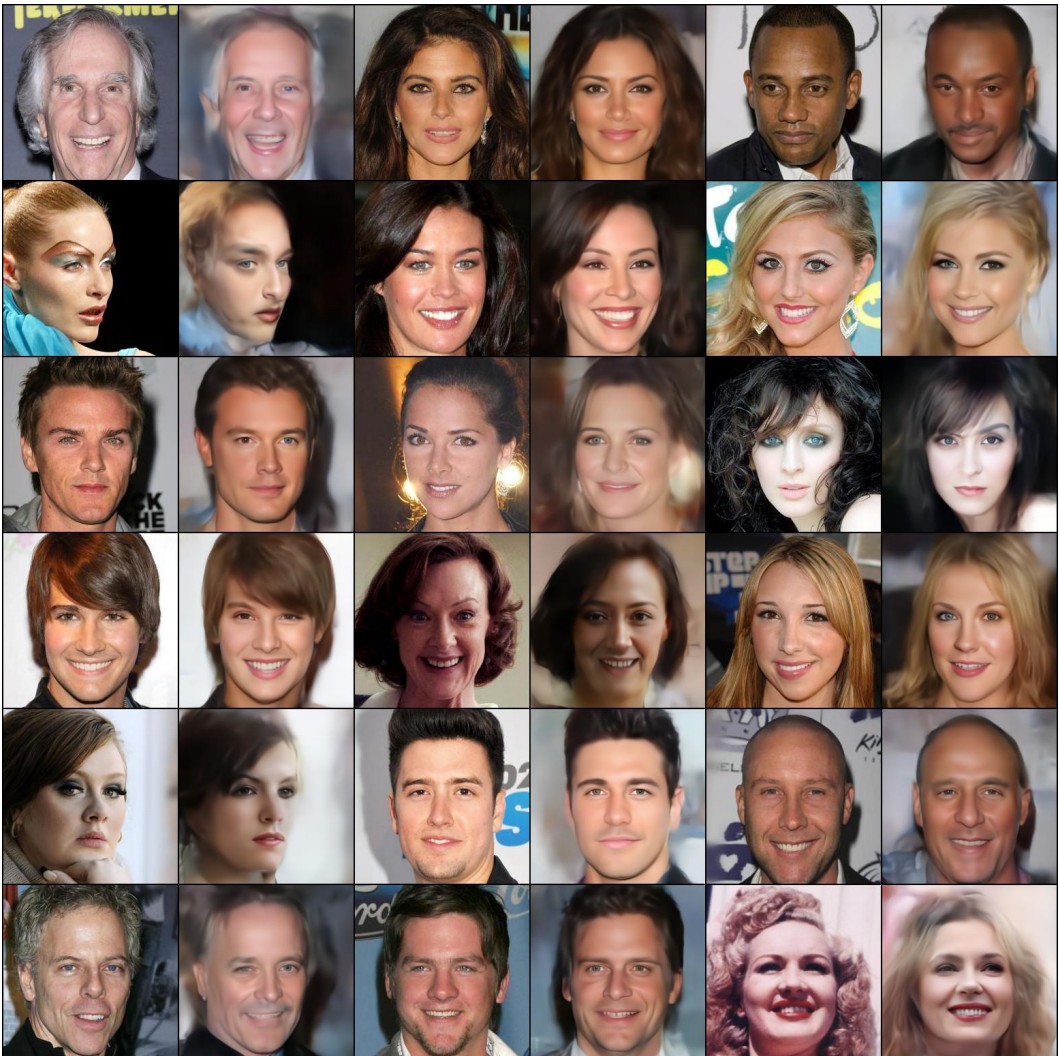

Figure 12: Image reconstructions by a model trained (as described in appendix A) on CelebA-HQ $256 \times 256$. For each pair of columns, the left is the original image, and the right is the reconstruction.

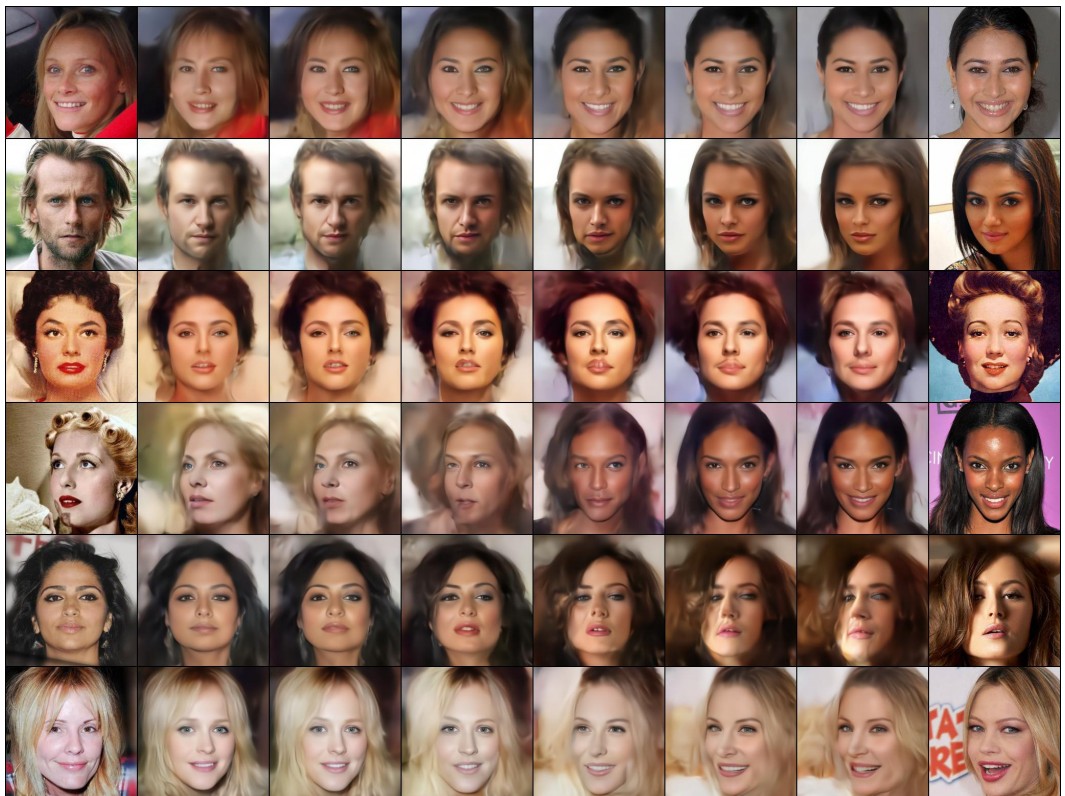

Figure 13: Latent space interpolations by a model trained (as described in appendix A) on CelebA-HQ $256 \times 256$.

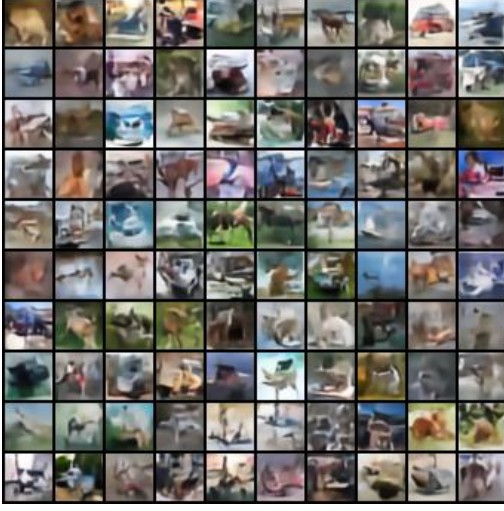

Figure 14: Randomly generated samples from the MoCA model trained on CIFAR-10 with FID 54.36 in table 1.

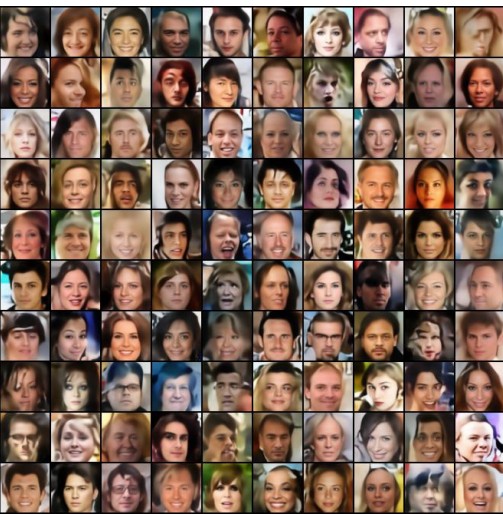

Figure 15: Randomly generated samples from the MoCA model trained on CelebA with FID 44.59 in table 1.

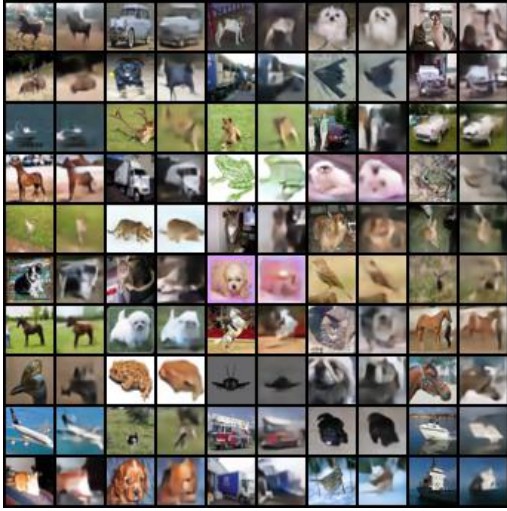

Figure 16: Reconstructed test samples from the MoCA model trained on CIFAR-10 with FID 54.36 in table 1.

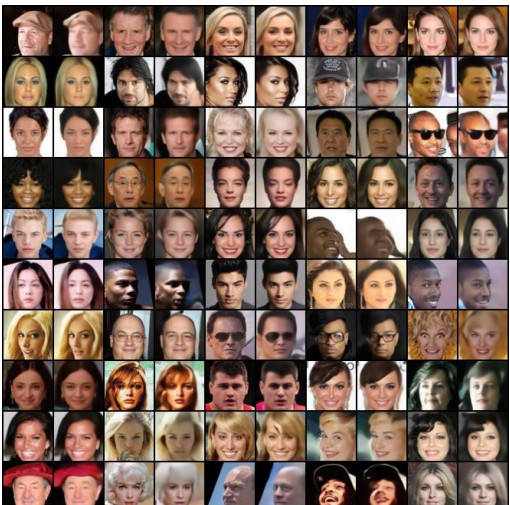

Figure 17: Reconstructed test samples from the MoCA model trained on CelebA with FID 44.59 in table 1.

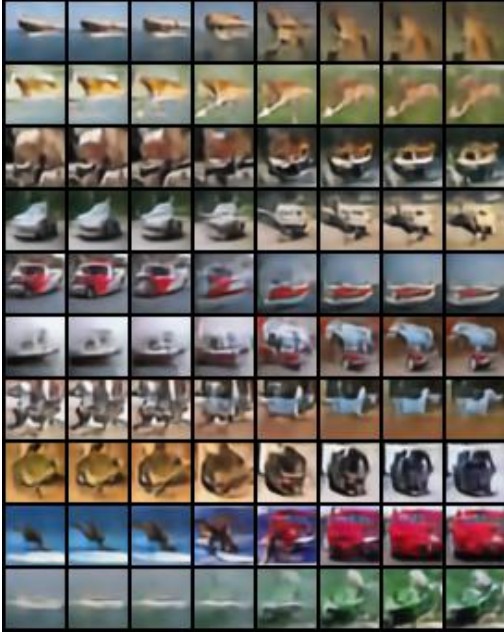

Figure 18: Interpolation between two test images in latent space for MoCA model trained on CIFAR-10 with FID 54.36 in table 1. The leftmost and rightmost columns are the original images from the corresponding test set.

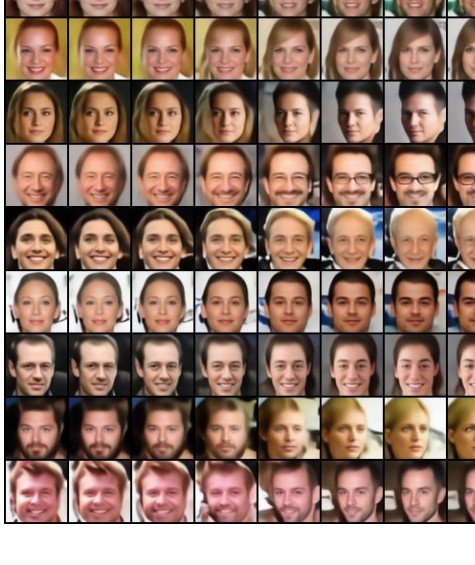

Figure 19: Interpolation between two test images in latent space for MoCA model trained on CelebA with FID 44.59 in table 1. The leftmost and rightmost columns are the original images from the corresponding test set.

