# OpenReview forum: "Momentum Contrastive Autoencoder: Using Contrastive Learning for Latent Space Distribution Matching in WAE"
_ICLR.cc/2022/Conference — ICLR 2022 Submitted_

### Official Review · Reviewer_nhFp · 2021-11-03

**Correctness:** 4
**Technical Novelty And Significance:** 3
**Empirical Novelty And Significance:** 3
**Recommendation:** 5
**Confidence:** 3

**Main Review:**

I notice this is a re-submission from ICLR-2021. Thus some of my comments are based on the differences between two versions.

## Strengths
1. The paper is well written and well motivated.
2. I think the idea of using contrastive learning to enforce a hypersphere prior for WAE is clever and neat.
3. The authors provide extensive ablations on hyperparameters.


## Weaknesses
1. My main concern is the performance of the proposed method on CIFAR10 and CelebA. The interpolation, reconstruction, and samples in Figure 6 are very blurry, and hard to justify the benefit of using the proposed approach. The reported FID in Table 1 and 2 are very high. It would be nice to include a comparison of [1] (which has FID of 5.25 and 24.08 on CelebA and CIFAR10 respectively). Also, why is the two-stage VAE baseline in the previous version removed?
2. It would be nice to include WAE-GAN in Figure 1 and 2, since it outperforms the proposed MoCA in Table 1.
3. I think it would be interesting to see how to integrate the instance contrastive loss as in DC-VAE [2] into the proposed MoCA.


[1] Aneja, Jyoti, et al. "Ncp-vae: Variational autoencoders with noise contrastive priors." arXiv preprint arXiv:2010.02917 (2020).
[2] Parmar, Gaurav, et al. "Dual contradistinctive generative autoencoder." Proceedings of the IEEE/CVF Conference on Computer Vision and Pattern Recognition. 2021.

**Summary Of The Paper:**

This paper proposes a new approach to train Wasserstein auto-encoders (WAE) with contrastive learning techniques. Specifically, the paper proposes to enforce the marginal matching constraint of WAE by exploiting the fact that contrastive learning objectives optimize the latent space distribution to be uniform over the unit hypersphere.

**Summary Of The Review:**

The main idea of the paper is well motivated. However, I still find the results on image datasets such as CIFAR10 and CelebA hard to justify the superiority of the proposed method over baselines. I lean towards weak rejection but am willing to amend my score if my concerns are addressed.

---

> ### Author Response · Authors · 2021-11-19
> **Rebuttal**
>
> Thank you for your feedback and comments.
>
> > **I notice this is a re-submission from ICLR-2021**: We sincerely hope that our paper is evaluated based on our claims and contributions of the current version, and the decision is not biased because of the last year's conference submission.
>
> > **The reported FID in Table 1 and 2 are very high. It would be nice to include a comparison of [1] (which has FID of 5.25 and 24.08 on CelebA and CIFAR10 respectively)**: It is unfair to compare the performance (both qualitative and quantitative) of our method with non-Wasserstein autoencoder methods. Our central contribution is to propose an algorithm for Wasserstein autoencoder (WAE) that achieves better optimization (stability and convergence speed) and establishing a novel connection between the WAE objective and how it can algorithmically be achieved using the Contrastive learning objective. As a result, the performance of our algorithm is theoretically bounded by the performance that can be achieved by WAE. We have conducted several experiments to verify our main claim directly in Fig 1, 2 and 3 in the main paper and additional experiments in rebuttal.pdf (supplementary material). In these experiments we measured the Wasserstein distance between the latent space marginal distribution and the prior distribution and compared the convergence speed with existing methods, and showed that our method achieves faster convergence speed.
>
> > **why is the two-stage VAE baseline in the previous version removed?**: We removed it because it was not related to Wasserstein autoencoder (WAE) and was a distraction. We re-wrote the experimental section to focus only on existing algorithms that implement the WAE objective because all our claims pertain to WAEs.
>
> > **It would be nice to include WAE-GAN in Figure 1 and 2**: Please see the last paragraph in the rebuttal to reviewer pPeS.
>
> > **I think it would be interesting to see how to integrate the instance contrastive loss as in DC-VAE [2] into the proposed MoCA**: We thank the reviewer for the suggestion, but this is beyond the scope of our paper since the contribution of DC-VAE is completely different from ours. There are probably many ways in which we can improve the performance of WAEs similar to how recent research has shown that the performance of VAEs can be improved significantly (e.g. NVAE and DC-VAE). However, our paper focuses on the core optimization challenge of WAE: matching the marginal distribution of the latent space to a prior distribution. We hope this makes sense.
>
> [1] Aneja, Jyoti, et al. "Ncp-vae: Variational autoencoders with noise contrastive priors." arXiv preprint arXiv:2010.02917 (2020).
>
> [2] Parmar, Gaurav, et al. "Dual contradistinctive generative autoencoder." Proceedings of the IEEE/CVF Conference on Computer Vision and Pattern Recognition. 2021.

---

### Official Review · Reviewer_ZJcJ · 2021-11-04

**Correctness:** 4
**Technical Novelty And Significance:** 3
**Empirical Novelty And Significance:** 2
**Recommendation:** 6
**Confidence:** 4

**Main Review:**

Strengths:

1. The paper is well-written and easy to follow. I do think that some of the notation such as "push forward" from measure theory, is really not needed or particularly useful here. Simpler terminology such as just using encoding and decoding functions would be more than sufficient.

2. Some of the experiments are interesting and show the effects of the proposed regularization e.g. on the singular value distribution of the latent representation.

3. Using a contrastive approach is a potentially effective way to match the prior and posterior distributions.

Weaknesses: It is unclear that the proposed regularizer results in qualitatively better reconstructions than the baselines. FID is not a perfect measure and the samples from baselines should be shown side-by-side with the proposed approach to know whether there is indeed an improvement. I found the CIFAR-10 reconstruction results are somewhat poor.

Question: the value of the lambda parameter is very large - what are the relative loss values during training/convergence (the reconstruction loss vs. regularizer loss)?

**Summary Of The Paper:**

The paper presents a regularization technique for Wasserstein Auto-Encoders, based on contrastive learning.

**Summary Of The Review:**

The paper is interesting but has some shortcomings. I would like to see some results of the baselines to decide if the proposed regularizer does indeed improve results qualitatively. I do not believe that FID is a proper measure of quality (not just for this paper but for measurement of GAN sample quality, in general). I give the paper a slightly positive score based on the idea, but I am looking forward to some samples in the rebuttal to decide my final score.

---

> ### Author Response · Authors · 2021-11-19
> **Rebuttal**
>
> We thank you for your comments and concerns. We have tried to address them below.
>
> > **notation such as "push forward" from measure theory, is really not needed**: This is a standard terminology and we were hoping it would make it easier for the readers to understand the content, but we are happy to remove it if it is not the case.
>
> > **FID is not a perfect measure and the samples from baselines should be shown side-by-side with the proposed approach**: Indeed FID is not a perfect measure to assess how well the objective of minimizing the divergence between generated and real samples has been achieved. Since our main claim in this paper was algorithmic, i.e., to propose an algorithm for minimizing the WAE objective that achieves faster and more stable convergence compared to existing non-adversarial WAE algorithms, we believe that the most direct way to verify our claim is to measure the divergence between the real and generated image distributions during the training process. We used the sliced Wasserstein distance (SWD) as a measure in our experiments and showed these experiments in Fig 1 and 2 in the paper both in the image space and the latent space and verified our claim to be true. This is a much more direct measure than FID score. We have also conducted additional experiments on synthetic dataset and CIFAR-10 in the supplementary material rebuttal.pdf in Fig 8 and Fig 9 (respectively) with multiple seeds and our conclusions remain the same. We believe that a higher weightage should be given to these experiments in our paper for a fair evaluation. Nonetheless, we also ran additional experiments using WAE-MMD (from the WAE paper using their github code) with multiple configurations and generated images from the model learned. We have shown these images along with images generated by our proposed method in Fig1-6 in the supplementary material file rebuttal.pdf.
>
> > **I found the CIFAR-10 reconstruction results are somewhat poor.**: WAEs do not generate images as sharp as GANs, at least not their current form. Perhaps modifications similar to those done by recent papers for VAEs (e.g. NVAE) are needed for WAE to improve the quality of images generated. Since our contribution was an algorithm aimed at improving the stability and convergence of WAE objective optimization, we believe it would be unfair to judge our work based on sample/reconstruction quality compared with SoTA methods or even other existing methods not related to Wasserstein autoencoders.
>
> > **the value of the lambda parameter is very large - what are the relative loss values during training/convergence**: We showed the final reconstruction loss at different values the regularization coefficient lambda in Fig 8 in appendix. We find that the reconstruction loss reduces more with larger values of lambda. To make it clearer, it should noted that the regularization term in WAE objective does not create a trade-off with the reconstruction loss. This is because the regularization term requires the marginal distribution of the latent space to match a prior, which does not conflict with minimizing the reconstruction error. This is not the case, for instance in VAE, where the posterior distribution of the latent space conditioned on each input is required to match the same prior (which does conflict with the reconstruction term).

---

### Official Review · Reviewer_pPeS · 2021-11-04

**Correctness:** 3
**Technical Novelty And Significance:** 2
**Empirical Novelty And Significance:** 2
**Recommendation:** 5
**Confidence:** 4

**Main Review:**

Strengths:
- Overall, this paper is well organized.
- The method proposed in this paper is a reasonable combination of existing state-of-the-art methods.
- Experimental results show that the proposed method has stability and faster convergence, which is promising.

Weaknesses:
- The authors claim that MoCA can generate images with high quality. However, the experimental results in this paper do not show this very well. First of all, although the authors claim that the results in Figure 5 and Figure 6 are "look realistic", some of the face images seem to be collapsed, and the interpolation between the two images seems to be discontinuous. Since there is no qualitative comparison with existing methods, we cannot judge these methods as "realistic". In Table 1, the authors show the quantitative comparison results with the existing methods, but there are some puzzling points. First, why does MoCA-A2, which has fewer parameters, have higher performance? Also, why does WAE-GAN perform better than MoCA? Since Table 1 shows that WAE-GAN is better than MoCA, shouldn't it be compared with WAE-MMD? Furthermore, why are the quantitative results of CelebA-HQ not shown?
- The authors show in their experiments that MoCA achieves faster convergence than existing methods, but they do not fully explain why MoCA shows such convergence. Why does the convergence of the proposed method become better when contrastive learning is included? In addition, Figure 1 shows the line graphs of only one training trial for each method, and the variance of each method is not shown. Therefore, I cannot judge whether the difference in results between methods is large or small.
- In Section 4.1, the authors compare WAE-MMD with MoCA as the original WAE algorithm, but I do not understand why they do not compare it with WAE-GAN, which is also original. I think this should be done because Table 1 shows that WAE-GAN has better image generation performance than the proposed method.
 For example, the authors employ MoCo, but how effective is this in improving the performance of the proposed method? The authors do not seem to have verified such a thing.

Minor comments:
 - The significant figures of the results of each method in Table 1 should be the same.
 - Section3: any fixed t -> any fixed \tau

**Summary Of The Paper:**

In this paper, the authors propose to use contrastive learning for matching in latent space in the Wasserstein autoencoder (WAE). In addition, they employ techniques such as momentum contrast in contrastive learning. Experimental results show that the proposed method, MoCA, is more stable and converges faster than existing methods. It is also capable of generating high-resolution images such as CelebA-HQ.


**Summary Of The Review:**

In terms of convergence and stability, the proposed method is considered to be effective to a certain extent. Also, the idea of using contrastive learning for WAE is interesting. However, the explanation of the claim and the presentation of the results are insufficient.

---

> ### Author Response · Authors · 2021-11-19
> **Rebuttal**
>
> Thank you for your detailed comments and suggestions. We have tried to address them below.
>
> > **The authors claim that MoCA can generate images with high quality**
> We apologize for any confusion. We never claimed that our algorithm generates "high quality" images, and to further clarify, generating high quality images compared to existing SoTA methods was not our goal. Our contribution centers around proposing an algorithm that is able to minimize the WAE objective better in terms of optimization (stability and convergence speed), and establishing a novel connection between the WAE objective and how it can algorithmically be achieved using the Contrastive learning objective.
>
> > **qualitative comparison with existing methods**
> We ran additional experiments using WAE-MMD (from the WAE paper using their github code) with multiple configurations and generated images from the model learned. We have shown these images along with images generated by our method in Fig1-6 in the supplementary material file rebuttal.pdf.
>
> > **why does MoCA-A2, which has fewer parameters, have higher performance?**
> Architecture A1 is a convolutional architecture (same as the one in the WAE paper for a direct comparison), while A2 is a ResNet architecture. We believe A2 outperforms A1, despite A2 having a smaller number of parameters, due to residual connections in A2. It is well known that ResNets outperform classical convolutional networks (CNN) in supervised learning. Even in generative modeling, ResNets outperform CNNs (cf Table 2 in [1]).
>
> > **why does WAE-GAN perform better than MoCA?**
> GANs can be challenging to train due to optimization instability, but when successfully trained, are known to generate sharp images. While WAE-GAN achieves an FID score of 42, which is indeed slightly better than 48.43 by our model (using the same architecture), we note that 1) our method's performance is closest to WAE-GAN; 2) this performance is better than all the currently existing non-adversarial methods for implementing WAE and autoencoders with hyper-spherical latent space.
>
> > **Furthermore, why are the quantitative results of CelebA-HQ not shown?**
> Existing papers on Wasserstein autoencoder algorithms (i.e. original WAE paper, sliced Wasserstein autoencoder, Sinhorn autoencoder, Cramer-Wold autoencoder, etc) mainly show quantitative results on CelebA, and some papers also show it on MNIST and CIFAR-10. Therefore, for a direct comparison with the results in their papers, we quantitatively show results on CelebA and CIFAR-10 and only show qualitative results on CelebA-HQ.
>
> > **Why does the convergence of the proposed method become better using contrastive learning?**
> We do not have a theoretical justification for why using the contrastive learning objective for implementing WAE converges faster than existing methods.
>
> > **Figure 1 shows the line graphs of only one training trial for each method, and the variance of each method is not shown**
> We conducted two additional sets of experiments. First, we re-ran the experiments shown in Fig 1 (on synthetic dataset) in our submission with 5 different seeds (as the reviewer suggested). The results are shown in Fig 8 of the supplementary material rebuttal.pdf. As can be seen, there is barely any impact of stochasticity in these experiments and the contrastive objective converges faster than existing methods and minimizes the sliced Wasserstein distance (tracked during experiment) to lower values. Secondly, we also ran the same experiment, but with the CIFAR-10 dataset instead of the synthetic dataset used in the above experiment. These results are shown in Fig 9 in the supplementary material rebuttal.pdf. The contrastive method for matching the latent space distribution with the prior distribution once again achieves the best convergence speed and lowest sliced Wasserstein distance between the marginal and the prior distributions.
>
> > **In Section 4.1, the authors compare WAE-MMD with MoCA... they do not compare it with WAE-GAN**
> In the paper we only ran experiments on existing algorithms that had non-adversarial objective. Based on the reviewer's suggestion, we ran experiments on both the synthetic task and CIFAR-10 (similar to section 4.1) using the GAN objective. We tried different variations of GANs (WGAN and the original GAN, and with weight clipping, gradient penalty, and spectral normalization) and tried to tune various hyper-parameters (learning rates of generator and discriminator, and frequency of discriminator updates). However, we could not get the SWD measured on both the datasets to reach the levels that the contrastive method reaches, and it can also be seen that the GAN loss is more unstable and has high variance across runs compared to all non-adversarial methods. These experiments are shown in Fig 10 and 11 of rebuttal.pdf in supplementary material.
>
> [1] Miyato, Takeru, et al. "Spectral normalization for generative adversarial networks." arXiv preprint arXiv:1802.05957 (2018).

---

### Decision · Program_Chairs · 2022-01-20

**Decision:**

Reject

**Comment:**

This paper presents a variant of the WAE which uses a contrastive criterion to enforce the marginal distribution matching constraint. Experiments show faster convergence in terms of Wasserstein distance, more visually appealing samples, and better FID scores compared with other WAE models.

The original WAE framework leaves open the choice of approximation for enforcing marginal distribution matching, and the original paper gives two such algorithms. Therefore, it's pretty natural to replace this approximation with something else (such as the contrastive criterion used here), so a submission would need to show evidence that it's significantly better than other approaches. Reviewers have expressed various concerns about the experiments. None of them are major problems, but overall the method doesn't seem consistently better than other WAE methods; e.g., the FID score is worse than that of WAE-GAN.

I encourage the authors to take the reviewers' comments into account in preparing the submission for future cycles.